# Trauma Exposure in Relation to the Content of Mother-Child Emotional Conversations and Quality of Interaction

**DOI:** 10.3390/ijerph16050805

**Published:** 2019-03-05

**Authors:** Mathilde M. Overbeek, Nina Koren-Karie, Adi Erez Ben-Haim, J. Clasien de Schipper, Patricia D. Dreier Gligoor, Carlo Schuengel

**Affiliations:** 1Section of Clinical Child and Family Studies, Vrije Universiteit, 1081 BT Amsterdam, The Netherlands; j.c.de.schipper@vu.nl (J.C.d.S.); p.d.dreiergligoor@student.vu.nl (P.D.D.G.); c.schuengel@vu.nl (C.S.); 2Amsterdam Public Health Research Institute, 1081 BT Amsterdam, The Netherlands; 3School of Social Work, University of Haifa, Haifa 3498838, Israel; nkoren@psy.haifa.ac.il (N.K.-K.); adi.ee1983@gmail.com (A.E.B.-H.)

**Keywords:** emotion conversation, emotion dialogue, mother-child interaction, parent-child communication, trauma exposure, marital violence, sexual abuse

## Abstract

Parent-child conversations contribute to understanding and regulating children’s emotions. Similarities and differences in discussed topics, quality of interaction and coherence/elaboration in mother-child conversations about emotional experiences of the child were studied in dyads who had been exposed to interpersonal trauma (N = 213) and non-trauma-exposed dyads (N = 86). Results showed that in conversations about negative emotions, trauma-exposed children more often discussed trauma topics and focused less on relationship topics than non-trauma-exposed children. Trauma-exposed dyads found it more difficult to come up with a story. The most common topics chosen by dyads to discuss for each emotion were mostly similar between trauma-exposed dyads and non-trauma-exposed dyads. Dyads exposed to interpersonal traumatic events showed lower quality of interaction and less coherence/elaboration than dyads who had not experienced traumatic events. Discussion of traumatic topics was associated with lower quality of mother-child interaction and less coherent dialogues. In conclusion, the effect of the trauma is seen at several levels in mother-child interaction: topics, behavior and coherence. A focus on support in developing a secure relationship after trauma may be important for intervention.

## 1. Introduction

Parents have a crucial role in helping children understand their inner world and supporting them in emotion regulation [1,2]. During daily conversations between parents and children, children learn which feelings are accepted and which are not. They learn if there are emotions that should be dampened and others that should be intensified. In addition, children discover whether there are emotional subjects to which their parents are attentive compared to others to which they are not. During these conversations children also learn whether they can rely on their parent to assist them through the reconstruction of a sequence of events in a meaningful and coherent manner. They gain an understanding of whether their parent is able to help them cope with negative feelings and regain a sense of confidence and security, particularly when recalling past aversive or traumatic events [3,4]. Parents who are capable of providing an enabling, child-focused, and organized emotional climate serve as a psychological secure base from which children can freely explore their emotional inner world. Inattentive and non-accepting parents on the other hand may negatively affect their child’s ability to create coherent and meaningful narratives about their emotional experiences [1,5].

Parental traumatic experiences may impact parents’ ability to support their children [6] and influence children’s emotion conversations with their parents. Emotion conversations between parents and children help children to understand their emotions, both regarding everyday events, as well as regarding highly significant and emotionally laden events. Meaning making regarding these events is important for a healthy emotional development [7]. Meaning making may be of particular importance for very stressful and emotion-laden traumatic events, such as child exposure to domestic violence or sexual abuse [8].

Co-constructed parent-child emotion conversations can be analyzed on three levels: (1) behavior of both partners, (2) coherence/elaboration of the conversation, and (3) content of the conversation [9,10]. The behavior of both partners describes the sensitivity of the caregiver and the cooperation of the child. Coherence refers to the narratives’ organization and describes the degree to which a story makes sense [8]. Coherent stories are internally consistent, fluent, detailed and focus on the emotion they ought to describe. As coherent stories are usually more detailed, they are likely to be more elaborate and consist of more words. The content of the conversation refers to what parent and child decide to discuss together. Most studies so far have focused on the way parent and child interact with each other and the coherence/elaboration of the narrative (e.g., [11,12]). The content of the conversation, particularly the topic parent and child decide to discuss together, has so far been largely neglected. The current study aims to fill this gap by focusing on all three aspects—the quality of mother-child interaction, coherence/elaboration of the narratives and the content of the stories—in mother-child conversations about different emotional experiences of dyads with and without trauma exposure. 

### 1.1. Quality of Interaction and Coherence/Elaboration in Narratives in Relation to Trauma Exposure

Exposure to traumatic events of either parent, child or both may hinder attunement between parent and child. This is unfortunate, because in highly stressful emotion-laden conversations maternal guidance is of great value for the child [13]. In families exposed to interpersonal traumatic events the parent-child interaction and coherence of the conversation are often of lower quality than in non-risk families. In addition, children with a less enabling parent likely feel more constrained to extensively explore and share their emotions and have less elaborative conversations compared to children with an enabling parent. Koren-Karie and colleagues [12] studied how sexually abused mothers talk with their non-abused children about emotional events. They found that the degree of maternal resolution of the trauma moderated the relationship between the abuse and the quality of mother-child interaction. More resolved mothers have been able to integrate the trauma in their view of the world and come to terms with the abuse, enabling them to focus more on the present and the current signals of their child. These mothers showed more sensitive guidance, their children were more cooperative, and their conversations were more coherent. Not only parental interpersonal trauma affects the quality of parent-child interaction. Additionally, when the child has been exposed to traumatic events, this influences mutual communication. Van Delft and colleagues [14] compared the quality of emotion conversations of sexually abused children with non-abused children and found a lower quality of interaction, less maternal sensitive guidance and less cooperation of the child. In a sample of mother-child dyads who both had been exposed to interpersonal trauma, in the case of exposure to marital violence, violence-exposed dyads showed less maternal sensitive guidance, less child cooperation and a lack of coherence/elaboration compared to a matched normative sample [15]. In addition, children showed striking behavior in interaction with their mothers after trauma exposure, such as overprotectiveness towards the parent and extreme aggressiveness [16].

Parental exposure to interpersonal trauma impacts well-being which in turn impacts parenting and mutual communication (e.g., [17,18]). Exposure to interpersonal trauma, such as marital violence or maternal childhood sexual abuse, has been shown to be associated with more aggressive parenting, less warmth, less consistency in parenting and less attention for the emotional needs of children [16,17]. Parents and children contribute differently to the emotion conversation and also influence each other both positively and negatively. Maternal acceptance and encouragement have been associated with children’s emotional openness [19] and cooperation [20]. By contrast, parents who were unsupportive or rejecting were less likely to have children willing to be emotionally open with their parents [19]. As parent-child conversations are based on reciprocity, quality of the conversation may be negatively influenced both when parents or children are exposed to traumatic events. Indirect evidence for this hypothesis comes from the findings of Van Delft and colleagues [14] and Koren-Karie and colleagues [12] on the effect of interpersonal traumatic experiences on emotion conversations. While in the study by Van Delft and colleagues only the children had been exposed to trauma and not the parents, in the study by Koren-Karie and colleagues only the mothers had been exposed to trauma and not the children. In both studies the quality of interaction and coherence of conversation was of lower quality in families which were highly impacted by the interpersonal traumatic events. 

### 1.2. Content of Emotion Conversations after Exposure to Interpersonal Trauma

Little research attention has been paid towards the content in parent-child emotion conversations. Grych and colleagues [21] found that children exposed to interparental violence had fewer positive representations of parent figures in a narrative task than control children. Shields and colleagues [22] found that maltreated children’s parental representations in a narrative task were less positive and more negative than non-maltreated children’s representations. Both studies used the narrative of the children and did not make use of a co-constructed parent-child narrative.

Two studies looked into references to emotions during jointly parent-child conversations [23,24], both comparing securely attached children with insecurely attached children in a normative sample. However, in both studies the focus was on the frequency of discussion of positive and negative events and the chosen topics were not part of the studies. In sum, only the rather global distinction between positive and negative topics has been examined. We could find no studies examining content of conversations of dyads with and without trauma exposure. 

### 1.3. Association between Content and Quality of Interaction and Coherence/Elaboration

The discussed content of the conversation may be related to the quality of the mother-child interaction and coherence/elaboration of their conversation. In order to discuss negative emotional experiences or traumatic events in an open, regulated, and coherent manner, children need a sensitive parent who provides them with an accepting and containing atmosphere and creates a secure base from which they can safely explore and share their thoughts, feelings, and (when occurring) painful memories [10]. However, establishing such an open and enabling atmosphere may be difficult for both partners, especially after experiencing interpersonal traumatic events for either parent, child or both [12,14,15].

Coherence and quality of parent-child interaction have been associated with emotion understanding; elaborative coherent stories within a secure relationship contributed to more emotion understanding [2]. Children whose parents are better able to provide a secure base will have access to a wider range of topics compared to children whose parents have more difficulties with providing sensitive guidance of emotion conversations. These children will be able to describe more elaborative well-organized stories in collaboration with their parents. Finally, Laible and Thompson [24] found that insecure dyads have more difficulty discussing emotions and made less references to feelings than secure dyads. Attachment has been associated with quality of interaction [25], which has been associated with trauma exposure (e.g., [15]). This suggests that trauma-exposed dyads will have more difficulty to come up with a topic for an emotion conversation. 

### 1.4. Current Study and Objectives

Exposure to interpersonal trauma has been found to impact parenting and parent-child communication [17,18], resulting in lower quality of interaction and less coherent and elaborative dialogues in trauma-exposed dyads (e.g., [15]). So far, little research attention has been paid towards the content in parent-child emotion conversations and its relations with the quality of interaction and coherence/elaboration. The objectives of this study were to examine similarities and differences in content, quality of interaction and coherence/elaboration in mother-child conversations about emotional experiences of children in dyads who have been exposed to interpersonal trauma compared with non-trauma-exposed dyads.

### 1.5. Hypotheses

1. Dyads exposed to traumatic experiences will refer to these traumatic experiences when discussing both positive and negative emotions. In addition, trauma-exposed dyads will describe fewer positive stories and more negative stories about parental figures. Differences in other topics are explored as well. 

2. Trauma-exposed dyads are expected to discuss a smaller range of topics than non-trauma-exposed dyads, and will more often not be able to decide on a topic together.

3. In trauma-exposed dyads the mother-child interaction is expected to be of lower quality than in non-trauma exposed dyads. Dyads exposed to traumatic experiences will provide less coherent and less elaborative emotion stories than non-trauma exposed dyads.

4. Discussion of traumatic topics will be associated with lower quality of mother-child interaction and less coherent dialogues. No specific hypotheses were made regarding discussion of other topics and quality of interaction or coherence.

## 2. Materials and Methods

### 2.1. Participants

A total of 299 mother-child dyads from different samples recruited in the Netherlands and Israel were included. The samples were combined into a group of participants exposed to interpersonal trauma and a control group (see Table 1 for descriptive statistics). There were no differences in children’s age or gender between the two groups, but in the trauma group the years of maternal education was lower and participants from the Netherlands were overrepresented.

### 2.2. Design and Procedure

All participants participated, as part of other studies [12,14,15,26,27], in the Autobiographical Emotional Events Dialogue (AEED; [28]). For the current study, all transcripts were recoded for content analysis, and the originally coded assessments of quality of mother-child interaction and coherence were used. Coders were blind to group allocation (trauma/non-trauma) and different coders coded the content of conversations and quality of mother-child interaction/coherence. 

### 2.3. Measures and Variables

*Content of mother-child emotion conversation* was measured with a newly developed coding system [29] for content analysis of the Autobiographical Emotional Events Dialogue (AEED; [28]). In this task mother and child were asked to co-construct four separate stories about events in which the child felt happy, scared, angry, or sad. The dialogues were transcribed verbatim. The discussed event for each emotion was classified into mutually exclusive topics based on the essence of the story to avoid multicollinearity (e.g., trauma-related, parent positive, parent negative, peer-related) and dichotomously coded (no = 0, yes = 1). First, the first two authors read together and discussed 94 transcripts to define common topics. A codebook was composed based on these topics generalized over emotions. In the ensuing coding process, topics were added when the existing themes did not cover a discussed topic. A total of 20% of transcripts were double coded for reliability. Differences between coders were resolved through discussion. In the next stage, themes that described similar topics (e.g., stories about different family members besides the core family were combined into the category ‘other family members’) or were difficult to differentiate (e.g., some stories were about ‘parents’ instead of ‘mother’ or ‘father’ resulting in the combined category ‘parent-related’) were combined. Finally, dialogues on very rare topics were recoded—if possible—into a more common topic, resulting in the final set of topics for analyses presented in Table 2.

*Quality of mother-child interaction* was also assessed in the AEED [28]. Transcripts were coded with the AEED coding system and quality of mother-child interaction was scored on seven scales for the mother and seven parallel scales for the child. Rating of the transcripts was done by marking indicators for the various scales as they appear throughout the transcript and then assigning a score on each of the scales both on the frequency and strength of these indicators. The scales are: *Focus on the task* (Mother/child is focused on the child’s emotions with no shifts to irrelevant details or to mother’s own feelings); *Clear boundaries* (Mother/child keeps their appropriate roles: mother does not force her own ideas/emotions on the child or becomes overwhelmed by the child’s themes, and the child does not assume a parental role such as promising to protect the mother); *Acceptance and tolerance* (Mother/child enables the other to express a wide range of emotional themes without being defensive or judgmental); *Hostility* (Mother/child shows thematic hostility, anger or derogation); *Involvement and reciprocity* (Mother/child is positively engaged in the task and shows genuine interest in the stories); *Containment of negative feelings* (Stories with negative themes are ended with positive resolutions and an emphasis on the child’s coping abilities and strength); *Structuring* and *Elaboration* (Mother facilitates the child in narrating rich and coherent stories and child tells rich and detailed stories). Each scale was scored between 1 and 9, and a higher score represented more of the coded behavior (in all scales except for *Hostility* high scores reflect positive behaviors. Scores for *Hostility* were inverted for the sum scales). For example, a mother who mostly accepts her child’s ideas but there are one or two indices in which she responds with slight impatience would get a high score on the *Acceptance* scale, whereas repeated rejection of the child’s ideas will lead to a low score on this scale. In addition, a very strong marker of specific behavior can lead to a low score, for instance when a mother derogates her child (e.g., “What a stupid story! Is that your best example? Couldn’t you come up with a better example? Now sit still and behave like a normal boy!”). All maternal scales were summed in a measure for ‘maternal sensitive guidance’ and all child scales combined represented ‘child cooperation and exploration’. Coders were blind for group allocation (trauma/non-trauma). Coders were trained by the developer of the coding system (N. Koren-Karie) and established adequate reliability. Adequate inter-rater reliabilities were reached for the composite scores (ICCs ranging from 0.76 to 0.95 for maternal sensitive guidance, 0.65 to 0.95 for child cooperation). Maternal sensitive guidance ranged from 21 to 60 (*M* = 43.31, *SD* = 7.37, possible range 9–63) and child cooperation and exploration ranged from 23.5 to 62 (*M* = 45.14, *SD* = 6.63, possible range 9–63). In addition to the scales, striking features in mother-child communication, such as extreme aggressiveness, child parentification or maternal frightening behavior, were coded dichotomously (present = 1, absent = 0) per emotion.

*Coherence of the conversation* was assessed with two overarching scales of the AEED coding system: *Adequacy of the story* (Mother and child describe separate stories for all four emotions and the stories match the emotions or themes they ought to describe); and *Coherence* (Mother and child construct stories that are fluent and clear). These scales are also scored between 1 and 9. High scores on these scales are given when mother and child construct stories that are internally consistent, fluent, detailed and match the emotions they ought to describe. The stories have a beginning, middle and end. The coder understands what the described event was and what the child was thinking, feeling and doing during that event. In coherent narratives there are no shifts to irrelevant topics or details. Negative emotions are not left open, but rather the stories end happily when possible or with an emphasis on the child’s strength and ability to competently deal with negative emotions. Inter-rater reliability for this composite score ranged from 0.73 to 0.93. Quality of the dialogue ranged from 2 to 17 (*M* = 8.69, *SD* = 3.65, possible range 2–18). 

### 2.4. Analyses

For all analyses SPSS (version 25) was used. Winsorizing was used to replace one outlier for maternal sensitive guidance. A total of 14 missings on maternal years of education were imputed with the total mean for analyses, but not for descriptive purposes. Differences in content of emotion dialogues between families with and without trauma are described with frequency analyses and tested for significance with chi-square tests. Differences in quality of mother-child interaction were tested with *t*-tests. The association between the content of the emotion conversations and quality of mother-child interaction and coherence are described using regression analyses, while controlling for children’s age and maternal years of education. 

## 3. Results

### 3.1. Difference in Content of Emotion Conversation between Dyads with and without Trauma

Topic choices of dyads who had been exposed to interpersonal trauma and dyads who had not been exposed to trauma were compared. As expected, in frequency analyses differences were found between discussed topics in dyads. For all negative emotions, trauma (e.g., incidents of parents fighting) was discussed more often in trauma-exposed dyads. Contrary to expectation, in conversations about the emotion happy, trauma-exposed dyads were more likely to discuss positive stories about parental figures than dyads without trauma. Overall, however, trauma-exposed dyads focused less on relationships (see Table 2 for an overview of topics per emotion between groups). Only for the emotion sad were trauma-exposed dyads less likely to come up with a story than non-trauma exposed dyads.

Interestingly, the most common topics chosen by dyads to discuss for each emotion were similar between families with and without exposure to interpersonal trauma, except for the discussion of trauma for scared in trauma-exposed dyads (Table 3).

### 3.2. Quality of Mother-Child Interaction

A description of the quality of mother-child interaction in samples with and without trauma exposure is provided in Table 4.

Mothers of trauma-exposed dyads were less sensitive in interaction with their children (t_(296)_ = 5.18, *p* < 0.001) and children were less cooperative (t_(296)_ = 4.39, *p* < 0.001). Exploratory analyses did not find significant differences within the trauma group between dyads where only mothers, only children, or both partners had been exposed (maternal sensitivity: mother/child trauma vs. both trauma, resp. t_(358)_ = −0.28, *p* = 0.780, t_(343)_= −0.07, *p* = 0.947; child cooperation: resp. t_(358)_ = −0.05, *p* = 0.963, t_(343)_ = −0.56, *p* = 0.576).

The additional questions regarding striking features in mother-child interaction showed the mothers in the trauma group to be more self-involved: they focused more on their own emotions (happy: t_(280.2)_ = −3.65, *p* < 0.001, scared: t_(235.6)_= −2.61, *p* = 0.010), more often rejected the topic of the child (scared: t_(212)_ = −2.88, *p* = 0.004), and more often told the story of the child by themselves (happy: t_(291.1)_ = −2.28, *p* = 0.024). 

### 3.3. Coherence and Elaboration of Conversation

In trauma-exposed families the dialogues were less coherent (t_(137.2)_ = 3.83, *p* < 0.001). Coherence was not worse in samples in which both partners had been exposed to trauma compared to samples in which only one partner had been exposed to trauma (mother/child trauma vs. both trauma, resp. t_(358)_= 0.43, *p* = 0.670, t_(343)_= −0.33, *p* = 0.739). See Table 4 for a description of coherence of emotion conversation in samples with and without trauma exposure.

Mothers of trauma-exposed dyads elaborated less while describing an emotional event than mothers of dyads in which no partner had been exposed to trauma (happy: F_(1, 297)_ = 9.03, *p* = 0.003, scared: F_(1, 297)_ = 24.97, *p* < 0.001, angry: F_(1, 297)_ = 12.25, *p* = 0.001, sad: F_(1, 297)_ = 20.38, *p* < 0.001). Similar results were found for children of trauma-exposed dyads (happy: F_(1, 297)_ = 5.87, *p* = 0.016, scared: F_(1, 297)_ = 17.76, *p* < 0.001, angry: F_(1, 297)_ = 7.78, *p* = 0.006, sad: F_(1, 297)_ = 5.36, *p* = 0.021) (see Table 5).

### 3.4. Associations between Content of Emotion Conversations and Quality of Mother-Child Interaction and Coherence

Controlled for children’s age and maternal years of education, topics of emotion conversation were regressed on quality of mother-child interaction and coherence. To limit the number of analyses, composite scores for relationship-related topics (stories on peers and family combined, for happy positive themes and for scared, angry and sad negative themes) and object-related topics were calculated for all four emotions, and regressed together with the topic ‘trauma’ and ‘no story’ (coded when the dyad was not able to come up with a story to discuss). Discussion of the topic ‘trauma’ was negatively associated with maternal sensitive behavior (scared: β = −0.19, *p* = 0.001; angry: β= −0.25, *p* < 0.001), child cooperation (scared: β = −0.13, *p* = 0.027, angry: β = −0.12, *p* = 0.038) and coherence (scared: β = −0.16, *p* = 0.008). 

‘No story’ was negatively associated with quality of mother-child interaction and coherence for almost all emotions (maternal sensitive behavior: happy: β = −0.12, *p* = 0.047, scared: β = −0.14, *p* = 0.014, angry: β = −0.12, *p* = 0.043; child cooperation: happy: β = −0.23, *p* < 0.001, scared: β = −0.13, *p* = 0.026, sad: β = −0.17, *p* = 0.003; coherence: happy: β = −0.18, *p* = 0.002, scared: β = −0.21, *p* < 0.001, angry: β = −0.13, *p* = 0.028, sad: β = −0.19, *p* = 0.001). 

Discussion of negative relationship-related topics for the emotion angry was negatively associated with maternal sensitive behavior (β = −0.13, *p* = 0.038). Additional analyses showed that particularly discussion of negative stories about parents (β = −0.17, *p* = 0.007), and not about siblings (β = −0.05, *p* = 0.405) or peers (β = −0.02, *p* = 0.718), was associated with lower maternal sensitive guidance. No other associations between the composite scores for relationship-related topics and object-related topics for the other emotions and maternal sensitive behavior, child cooperation or coherence were found. 

Trauma-related topics were assumed to be more difficult to discuss than daily problems such as a fight with a classmate or a bad grade, placing dyads who have been exposed to trauma at higher risk for negative mother-child interaction. However, additional analyses within the sample of trauma-exposed dyads showed there were no differences in maternal sensitive guidance (t_(210)_ = 1.22, *p* = 0.226), child cooperation (t_(210)_ = 1.34, *p* = 0.183) or coherence (t_(210)_ = 0.85, *p* = 0.397) between the dyads who discussed a topic coded under ‘trauma’ and dyads who did not discuss a traumatic topic. Hierarchical regression analyses with the control variables added in the first step, sample allocation in the second step, and the topics (trauma, no story, relationship-related, object-related) in the third step showed similar results. Exposure to trauma significantly predicted maternal sensitive behavior, child cooperation and coherence. Of the coded content only ‘no story’ was significantly associated with lower maternal sensitive behavior, child cooperation and coherence. 

## 4. Discussion

As hypothesized, trauma-exposed dyads more often discussed trauma-related topics, although the number of positive or negative stories about parental figures was not significantly different. The fact that children of trauma-exposed dyads chose the trauma as topic for discussion more often than non-trauma exposed children is not really surprising since this is a salient topic for both conversation partners. Contrary to expectation, trauma-exposed dyads did not describe fewer positive and more negative stories about parental figures. This finding has been found in two studies with abused children participating in a narrative task [21,22]. In the current study, the mutually exclusive coding of the topic of a story might have hindered coding more stories as negative about parental figures in trauma-exposed dyads. On the other hand, compared to earlier studies, mother and child participated together in a discussion task, which could make it more difficult for the child to discuss negative stories with the parent in which the parent is the focus of the negative emotions, especially when the mother-child interaction is of lower quality; and could be a sign of the child’s tendency to please the mother. Support for this interpretation can be found in the fact that for the emotion happy, trauma-exposed dyads were even more likely to discuss positive stories about parental figures. Overall, trauma-exposed dyads focused less on relationships in their conversations than non-trauma-exposed dyads. In light of previous research with traumatized individuals this can be explained by the detrimental effects of trauma exposure and posttraumatic stress symptoms, such as emotional numbing, on attachment behavior and interpersonal functioning. Traumatic memories can lead to distancing and avoidance of interpersonal triggers that create traumatic re-enactments [30]. To protect oneself from traumatic memories individuals rather focus on topics which are not relationship-related and therefore ‘safe’.

Contrary to our hypothesis, dyads exposed to interpersonal trauma did not discuss a smaller range of topics than non-trauma-exposed dyads. However, they did have more trouble deciding on a topic to discuss together and were less likely to come up with a story than non-trauma exposed dyads. The similar range of topics discussed by trauma-exposed and non-trauma-exposed dyads shows that, despite trauma, on a daily basis children are still mainly consumed by worries about friends, sports or common scary things such as thunderstorms. 

As expected, dyads exposed to traumatic events showed lower quality of interaction than dyads who had not experienced traumatic events, and their conversations were less coherent and elaborate. These findings have been found in previous studies (e.g., [15]). In addition, mothers in the trauma samples were more self-focused and more often rejected the child’s story, which may be related to their higher probability to not be able to decide on a topic together (Hypothesis 2). These findings are in line with previous research showing that traumatized mothers have greater difficulty with parenting [31] and children need their parents as a psychological secure base from which to explore their emotional inner world [1,5]. Additional explorative analyses showed that the quality of mother-child interaction was similar in samples in which either one (child/mother) or both partners had been exposed to traumatic experiences. Although this result needs to be interpreted with caution, because of small sample sizes of the samples in which only one partner had been exposed to trauma, this is an interesting finding. One would expect that mothers, who are the mature and experienced partner of the dyad, will have more impact on the conversation than the child, which has been found before [32]. Trauma-exposure however may have such a great influence that the dyad as a whole is negatively influenced, regardless of which partner was exposed to the traumatic events. Studies on the effect of sexual abuse of one’s child on parental functioning show similar results [33]. This stresses the vulnerability of the dyad, since their ability to create a meaningful, focused and coherent narrative together is negatively affected when one part of the dyad has suffered trauma. 

As hypothesized, discussion of traumatic topics was associated with lower quality of mother-child interaction and less coherent dialogues. This could not be explained by the fact that trauma-related topics may be more difficult to discuss than more daily problems, suggesting that it is particularly the exposure to trauma and not the discussion of the traumatic events which is associated with the lower quality of mother-child interaction. Discussion of relationship-related topics was not related to the quality of mother-child interaction, except for an association between more negative discussion about parents and lower quality of maternal behavior. This seems contrary to our other findings that trauma-exposed dyads have more difficulties discussing negative events about parents with parents. On the other hand, it may be simply the case that children of less sensitive mothers have more negative incidents between themselves and their mother to discuss.

### 4.1. Strengths and Limitations

To our knowledge this is the first study which looks into the association between the content of emotion dialogues and quality of mother-child interaction and coherence in dyads exposed to interpersonal trauma compared to dyads not exposed to trauma. Strengths of the study are the use of an observation measure for coding the quality of mother-child interaction, and the combination of different samples from previous studies enabling us to compare relatively large samples of families with and without exposure to interpersonal trauma. The samples with and without trauma exposure were comparable on all background variables except for maternal years of education and country of sampling. A limitation of the study is that both the content of the conversations and the quality of the interaction are measured with the same instrument (AEED). However, for both constructs different coding systems were used and were coded by different coders who were blind to group allocation. 

### 4.2. Future Directions

In the current study, only mothers and their children participated. Fathers interact differently with their children than mothers [34]. Most studies on the effect of trauma on parenting have been done with mothers, and little is known on how trauma-exposure affects parenting in fathers [35]. In future studies it would be interesting to look into how trauma affects emotion conversations between fathers and their children as well as emotion conversations within family systems of parents and siblings. Children may derive different forms of support from different persons, and may therefore benefit differently from conversations with their fathers and their mothers. More insight into these support networks of children may be helpful in the provision of treatment after trauma exposure. In addition, the analyses in the current study on (lack of) differences in quality of interaction between samples in which one partner or both partners had been exposed to trauma gave interesting food for thought, but were underpowered. A larger study with equal cells for trauma exposure of one (separately for mother and child) or both partners would help to clarify this point. 

## 5. Conclusions

Exposure to interpersonal trauma affects not only individual family members but also the parent-child dyad as a whole and its lasting effects are still noticeable many years later. The effect of the trauma is seen at several levels: sensitive guidance of the mothers, cooperation of the child, the ability of the dyad to form a coherent and elaborate story and the topic the dyad choses to talk about. In this study a link has been found between exposure to interpersonal traumatic events and emotion dialogues. Higher quality of parent-child interaction, including narratives, has been associated with better child functioning (e.g., [10]). Clinicians working with families exposed to interpersonal trauma may pursue problematic conversations about emotions as a possible explanation for children’s problems and might consider interventions that focus on this aspect of parent-child interaction. 

## Figures and Tables

**Table 1 ijerph-16-00805-t001:** Descriptive statistics.

	Both No Trauma	Trauma	Statistics
N (%)	86 (29)	213 (71)	-
Age child (M (SD), range)	10.2 (3.5), 4–18	9.7 (2.6), 4–18	t_(123.6)_ = 1.17, *p* = 0.24
Gender child (% boys)	44.2	51.2	χ^2^ = 1.20, *p* = 0.27
Years education mother (M (SD), range)	16.9 (3.1), 8–25	13.8 (3.8), 4–24	t_(283)_ = 6.50, *p* < 0.001
Nature trauma, N (%) ^a)^	N/A	Marital violence: 151 (70.9) Mother sexually abused: 32 (15.0) Child sexually abused: 17 (8.0) Mother and child sexually abused: 13 (6.1)	-
Participants from Netherlands, N (%)	57 (66)	181 (85)	χ^2^ = 13.19, *p* < 0.001

^a)^ Percentages based on the trauma group only.

**Table 2 ijerph-16-00805-t002:** Frequency of topics between dyads with and without interpersonal trauma exposure.

	Both No Trauma (N = 86)	Trauma (N = 213)	Χ^2^ (*p*)
**Happy**			
Trauma	0% (0)	4.2% (9)	3.75 (0.053)
Family relations (positive)	12.8% (11)	17,1% (38)	1.14 (0.286)
Family relations parents (positive)	4.7% (4) *	12.2% (26) *	3.87 (0.049)
Peers	25.6% (22) **	11,3% (24) **	9.64 (0.002)
Success	18.6% (16)	14.1% (30)	0.96 (0.327)
Fun activities (including birthday, holidays)	39.5% (34)	51.6% (110)	3.60 (0.058)
Get thing	17.4% (15)	27.2% (58)	3.18 (0.075)
No story	1.2% (1)	3.8% (8)	1.41 (0.235)
**Scared**			
Trauma	1.2% (1) **	13.1% (28) **	10.04 (0.002)
Family relations (positive)	8.1% (7) *	2.8% (6) *	4.17 (0.041)
Family relations parents (positive)	7.0% (6)	2.8% (6)	2.75 (0.097)
Family relations (negative)	5.8% (5)	8.5% (18)	0.60 (0.439)
Family relations parents (negative)	4.7% (4)	8.5% (18)	1.30 (0.255)
Peers	9.3% (8)	8.5% (18)	0.06 (0.813)
Illness/check-ups	8.1% (7)	12.7% (27)	1.25 (0.263)
Separation/death	7.0% (6)	2.8% (6)	2.75 (0.097)
Frights (including animals, natural forces, dark)	25.6% (22)	28.6% (61)	0.29 (0.593)
Scary movie (including nightmares)	20.9% (18)	22.5% (48)	0.09 (0.762)
Scary people	9.3% (8)	5.2% (11)	1.76 (0.184)
Adventures	5.8% (5)	5.6% (12)	0.00 (0.951)
No story	2.3% (2)	4.2% (9)	0.62 (0.430)
**Angry**			
Trauma	1.2% (1) *	8.0% (17) *	5.03 (0.025)
Family relations negative	44.2% (38) *	31.9% (68) *	4.03 (0.045)
Family relations parents (negative)	24.4% (21)	19.7% (42)	0.81 (0.367)
Family relations siblings (negative)	24.4% (21) *	13.6% (29) *	5.14 (0.023)
Peers	27.9% (24)	33.8% (72)	0.98 (0.323)
Not get way	14.0% (12)	17.8% (38)	0.67 (0.415)
Treated badly	8.1% (7)	10.3% (22)	0.34 (0.563)
Frustration	4.7% (4)	1.9% (4)	1.81 (0.179)
Object-related	5.8% (5)	3.8% (8)	0.62 (0.430)
No story	4.7% (4)	3.3% (7)	0.32 (0.570)
**Sad**			
Trauma	1.2% (1) ***	14.6% (31) ***	11.50 (<0.001)
Family relations (positive)	11.6% (10)	7.0% (15)	1.68 (0.195)
Family relations parents (positive)	7.0% (6)	6.1% (13)	0.08 (0.779)
Family relations (negative)	11.6% (10)	16.0% (34)	0.92 (0.338)
Family relations parents (negative)	7.0% (6)	12.2% (26)	1.75 (0.185)
Peers	10.5% (9)	21.1% (45)	3.68 (0.055)
Illness/accidents	18.6% (16)	17.4% (37)	0.06 (0.800)
Separation/death	32.6% (28) *	21.1% (45) *	4.34 (0.037)
Not get way	10.5% (9)	7.5% (16)	0.70 (0.404)
Object-related	4.7% (4)	3.8% (8)	0.13 (0.721)
Event-related	3.5% (3)	2.8% (6)	0.10 (0.758)
No story	0% (0) *	5.6% (12) *	5.05 (0.025)

* *p* ≤ 0.05; ** *p* ≤ 0.01; *** *p* ≤ 0.001.

**Table 3 ijerph-16-00805-t003:** Most common topics per emotion between dyads with and without trauma exposure.

Both No Trauma (N = 86)			Trauma (N = 213)			
		% (N of Stories)				% (N of Stories)
Happy			Happy			
**1**	Fun activities	39.5 (34)	**1**	Fun activities		51.6 (110)
**2**	Peers	25.6 (22)	**2**	Get thing		27.2 (58)
**3**	Success	18.6 (16)	**3**	Family relations (positive)		17.1 (38)
Scared			Scared			
**1**	Frights	25.6 (22)	**1**	Frights		28.6 (61)
**2**	Scary movie/ nightmare	20.9 (18)	**2**	Scary movie/ nightmare		22.5 (48)
**3**	Peers	9.3 (8)	**3**	Trauma		13.1 (28)
Angry			Angry			
**1**	Peers	27.9 (24)	**1**	Peers		33.8 (72)
**2**	Parents	24.4 (21)	**2**	Parents		19.7 (42)
**3**	Siblings	24.4 (21)	**3**	Not get way		17.8 (38)
Sad			Sad			
**1**	Death/separation	32.6 (28)	**1**		Death/separation	21.1 (45)
**2**	Illness/accidents	18.6 (16)	**2**		Peers	21.1 (45)
**3**	Family relations (pos/neg)	11.6 (10)/11.6 (10)	**3**		Illness/accidents	17.4 (37)

**Table 4 ijerph-16-00805-t004:** Quality of mother-child interaction in samples with and without trauma exposure.

	Both no Trauma (N = 86) *M* (*SD*)	Trauma (N = 213) *M* (*SD*)
Maternal sensitive guidance	46.64 (7.05)	41.96 (7.07)
Child cooperation	47.71 (6.55)	44.10 (6.39)
Coherence	10.02 (3.97)	8.16 (3.37)

**Table 5 ijerph-16-00805-t005:** Number of words of mothers and children used for describing an emotional event.

	Both No Trauma (N = 86) *M* (*SD*)	Trauma (N = 213) *M* (*SD*)
**Happy**		
Mother	41.57 (33.16)	28.69 (33.72)
Child	48.80 (49.27)	36.18 (36.84)
**Scared**		
Mother	65.06 (60.74)	36.59 (36.16)
Child	78.17 (76.12)	45.93 (51.97)
**Angry**		
Mother	61.69 (63.90)	38.76 (45.24)
Child	70.47 (78.88)	47.86 (56.05)
**Sad**		
Mother	61.95 (61.54)	33.71 (42.90)
Child	55.44 (50.92)	39.82 (53.58)

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
