# Peer review of "Trauma Exposure in Relation to the Content of Mother-Child Emotional Conversations and Quality of Interaction"

_ijerph, 2019, doi:10.3390/ijerph16050805_

Reviewer 1 Report

Abstract: “Trauma-exposed 18 dyads came up with a story with more difficulty.” – this could be interpreted to mean that the story they came up with was more complex or they found it more difficult to come up with a story. Please clarify.

When the authors refer to interparental violence it is not clear if they mean violence between the child’s parents or the mothers own parents or both?

“The discussed event for each 162 emotion was classified into mutually exclusive topics (e.g. trauma-related, parent positive, parent 163 negative, peer-related) “ Surely a topic could be trauma-related and parent negative so these are not mutually exclusive?

“(shift of focus, boundary dissolution, acceptance 176 and tolerance, hostility, involvement and reciprocity, closure and resolution of negative feelings, 177 structuring, and elaboration). Each scale was scored between 1 and 9, and a higher score represented 178 more of the coded behavior. All maternal scales were summed in a measure for ‘maternal sensitive guidance’ and all child scales combined represented ‘child cooperation and exploration’.” – Please be clear on which, if any, of the scales were reverse scored (for example hostility is bad while acceptance and tolerance is good so simply adding those scores together would result in a nonsensical score. Please also clarify if a high score is a good thing or a bad thing (“more of the coded behaviour” is ambiguous).

Analysis – would ANOVA have been more appropriate means of analysis for differences in mother-child interactions?

Grammar 336-337: Fathers interact different with 337 their children than mothers [34]

The conclusions are a little vague. I would like to hear more about avenues specific to the findings of this study – the aspects of mother-child communication that could be addressed in interventions. 

Author Response

Response to Reviewer 1

Trauma exposure in relation to the content of mother-child emotional conversations and quality of interaction, manuscript ID IJERPH-408374.

Thank you for your time and effort in reviewing our paper. Below we will address your concerns one by one.  

1) Abstract: “Trauma-exposed dyads came up with a story with more difficulty.” – this could be interpreted to mean that the story they came up with was more complex or they found it more difficult to come up with a story. Please clarify.

We agree with the reviewer that this sentence can be interpreted in different ways. We changed the sentence to be more in line with the results regarding the second hypothesis that dyads exposed to interpersonal traumatic events more often were not able to come up with a story (page 1: line 18).

2) When the authors refer to interparental violence it is not clear if they mean violence between the child’s parents or the mothers own parents or both?

Thank you for pointing out this unclear use of words. We rephrased the term ‘interparental violence’ to ‘marital violence’ in the sentences where it was unclear whether the use of ‘interparental violence’ referred to the child’s or mother’s parents, to make it more clear we mean to describe violence between the child’s parents (page 1: line 27, line 82-83, line 88, Table 1).

3) “The discussed event for each emotion was classified into mutually exclusive topics (e.g. trauma-related, parent positive, parent negative, peer-related)“. Surely a topic could be trauma-related and parent negative so these are not mutually exclusive?

We agree with the reviewer that a topic could be both related to a traumatic event as well as to a negative experience regarding the parent. However, to prevent multicollinearity, we focused in these cases on the essence of the story to be able to decide on one topic per story. For example, a child tells a story in which he was afraid when his father hit his mother and the police came to the house and his father was taken into custody. In this story, the child seems mostly focused on the line of events regarding the marital violence and this story would be coded under ‘trauma-related’. In another example in which the child tells a story in which he is angry at his father for hitting his mother and doesn’t want to see his father anymore, the story is clearly more on the relationship with the father, and would be coded under ‘parent negative’. Stories in which this coding was ambiguous were discussed between two or more coders. We added a sentence in the manuscript to explain this more clearly (page 5, line 176-177).

4) “(shift of focus, boundary dissolution, acceptance and tolerance, hostility, involvement and reciprocity, closure and resolution of negative feelings, structuring, and elaboration). Each scale was scored between 1 and 9, and a higher score represented more of the coded behavior. All maternal scales were summed in a measure for ‘maternal sensitive guidance’ and all child scales combined represented ‘child cooperation and exploration’.” – Please be clear on which, if any, of the scales were reverse scored (for example hostility is bad while acceptance and tolerance is good so simply adding those scores together would result in a nonsensical score. Please also clarify if a high score is a good thing or a bad thing (“more of the coded behavior” is ambiguous)

Thank you for pointing out this omission. In the manuscript, in line 188-208, more information is provided about the topic of each scale, the reversing of scores for Hostility and how composite scores for mothers and for children were computed.

5) Analysis – would ANOVA have been more appropriate means of analysis for differences in mother-child interactions?

Since we only compared two groups (trauma-exposed vs. non-trauma-exposed) we would argue that a t-test provides similar results to an ANOVA.

6) Grammar 336-337: Fathers interact different with their children than mothers [34]

Thank you for pointing out this language error, we have changed the sentence (line 380).

7) The conclusions are a little vague. I would like to hear more about avenues specific to the findings of this study – the aspects of mother-child communication that could be addressed in interventions.

We have rewritten the conclusion to be somewhat more specific regarding clinical implications (page 13: line 397-401). However, since this is a non-experimental study we are hesitant to draw clear clinical implications.

Reviewer 2 Report

Review of Manuscript: Trauma exposure in relation to the content of mother- child emotional conversations and quality of interaction.

The authors examined the differences in narrative coherence, elaboration and dyadic interaction between trauma-exposed and non-exposed mother-child dyads during joint storytelling about past experiences. The study found significant differences in these variables between the co-constructed narratives of trauma-exposed vs. non-exposed dyads.

The manuscript was a pleasure to read, and the findings were intriguing, adding another layer to research demonstrating the importance of co-constructed mother-child narratives in helping children to make sense of their experiences.

Below are a few concerns.

Introduction:

1.      The writing consists of many run-on sentences that can be difficult to keep track of at times, e.g., between lines 33 and 38 of the Introduction. The writing can seem a bit awkward in these kinds of sentences.

2.     Line 61 – sub-heading should read “in relation to”.

3.     There is an assumption made throughout the Introduction that some topics are familiar to readers, such as coherence and elaboration. These need to be more clearly defined in the Introduction. Related to this, much of the Introduction, focuses on the variable of Interaction between dyadic partners, with little discussion of the other variables that appeared to be important later on.

Method:

1.     The AEED coding system needs to be described in more detail. The subscales are simply listed and not defined or elaborated upon, including what each point in the 1-9 scale refers to. Examples from the narratives would also be very helpful in elucidating the scales to the read.

2.     Similarly, the coherence coding scheme needs to defined, and examples from the narratives of high and low ends of the scale should be included.

Discussion:

1.     The discussion should be more closely aligned with the hypotheses listed at the end of the Introduction and organized around those hypotheses so that it is easier to read and follow.

2.     Line 300 of the Discussion – the sentence is unclear.

3.     Line 336 – it should be “fathers interact differently”

Overall, I found the findings to be provocative and the study opens the door to many unanswered questions with regards to mother-child dyads who have experienced trauma.

Author Response

Response to Reviewer 2

Trauma exposure in relation to the content of mother-child emotional conversations and quality of interaction, manuscript ID IJERPH-408374.

Thank you for reviewing our paper and for your positive feedback on our study. Below we will address your concerns one by one.  

Introduction

1) The writing consists of many run-on sentences that can be difficult to keep track of at times, e.g., between lines 33 and 38 of the Introduction. The writing can seem a bit awkward in these kinds of sentences.

Thank you for pointing this out to us. We shortened or broke up the run-on sentences (page 1: line 33, line 36-37; page 2: line 47, line 64-65; page 3: line 110, line 128, line 141; page 12: line 341).

2) Line 61 – sub-heading should read “in relation to”.

Thank you very much for your attentive reading. The subheading has been changed accordingly.

3) There is an assumption made throughout the Introduction that some topics are familiar to readers, such as coherence and elaboration. These need to be more clearly defined in the Introduction. Related to this, much of the Introduction, focuses on the variable of Interaction between dyadic partners, with little discussion of the other variables that appeared to be important later on.

Coherence refers to the narratives’ organization and describes the degree to which a story is fluent and clear (Koren-Karie et al., 2000). Because coherent stories are usually more detailed, they are likely to be more elaborate and consist of more words. This additional explanation has been added to the Introduction (page 2, line 54, 56-57), and the intertwining of both constructs has been stated more clearly throughout the text (page 2: line 52, line 59, line 62, line 65, line 83; page 3, line 116, line 118, line 137, line 141). In addition, a sentence has been added describing that children in a secure relationship with their parent, with higher quality of parent-child interaction, describe more elaborative and coherent stories in collaboration with their parent (line 129-130).

The reviewer remarks correctly that most of the Introduction focuses on the differences in quality of interaction in samples with and without trauma exposure and less on differences in coherence and elaboration. This is because less research has been done on the constructs of coherence and elaboration than on the quality of parent-child interaction in trauma samples compared to control samples. For example, the study by Van Delft et al. (2018) only reported differences in quality of interaction between sexually abused children and matched controls, and provided no information on differences in coherence between both samples.

Methods

1) The AEED coding system needs to be described in more detail. The subscales are simply listed and not defined or elaborated upon, including what each point in the 1-9 scale refers to. Examples from the narratives would also be very helpful in elucidating the scales to the reader.

We thank the reviewer for this request for more information on the AEED coding system. The scales for mother and child are now described more extensively on page 8 line 188-201. In addition, we added the sentence that on all scales except for Hostility high scores reflect positive behaviors (line 200) and scores for Hostility were inverted for the sum scales. An example of interaction coded on the Acceptance-scale has been added to the manuscript on page 8 line 202-208 for further clarification.

2) Similarly, the coherence coding scheme needs to defined, and examples from the narratives of high and low ends of the scale should be included.

Thank you for this suggestion. A description of the two scales for coding coherence of the conversation has been added to the manuscript (page 8, line 219-227), as well as an explanation on what a coherent story entails.

Discussion
1) The discussion should be more closely aligned with the hypotheses listed at the end of the Introduction and organized around those hypotheses so that it is easier to read and follow.

We thank the reviewer for making this important point. We have integrated the hypotheses in the discussion to make the discussion easier to read and to follow (line 313-315; line 331-334; line 337-338; line 355-356). To make the discussion more readable hypothesis 2 and 3 have been rewritten as one hypothesis (Hypothesis 2, page 4, line 149-150), and hypothesis 4 and 5 have been combined in Hypothesis 3 (page 4, line 151-153).

2) Line 300 of the Discussion – the sentence is unclear.

In some instances mother and child were not able to decide on a topic together and were therefore unable to describe a story together, resulting in a coding of ‘no story’. We understand our choice of words in linking this outcome with more self-focused behavior of the mother has been unclear, and have therefore rewritten this sentence and made a referral to Hypothesis 2 in which this association between trauma-exposure and the ability to construct a story together has been described (page 12, line 341).

3)  Line 336 – it should be “fathers interact differently”

Thank you very much for pointing out this language error, we have changed the sentence (line 377).

Reviewer 3 Report

The manuscript addresses an interesting and important topic and potentially adds to previous literature. It also fits with the Journal aims and I think it can be published, should the authors be prepared to revise some minor points, as follows:

- in the introduction section, it is not always clear whether the impact on the quality of dialogues depends on parents' or offspring's traumatic experiences (or both). I suggest clarify this point;

- at line 71, the authors state "less resolved mothers...". In what sense "resolved"? Do the authors refer to an attachment theory-related resolution of loss or trauma, or do they refer to a psychoanalytic framework?

- I feel that the reader could benefit from some explanation of how different kinds of trauma (violence, sexual abuse, neglect, etc.) impact the quality of parent-child interaction and dialogues; and is there any difference in case of interpersonal trauma?

- I suggest adding specific objectives;

- It is not clear who administered the measures? Were the tools administered by the same professionals in the different studies the sample comes from?

- The analyses were well conducted and I do not have any suggestion;

- The result section is well written and clear;

- In the discussion section, I would suggest adding a brief separate paragraph on possible clinical implications.

Author Response

Response to Reviewer 3

Trauma exposure in relation to the content of mother-child emotional conversations and quality of interaction, manuscript ID IJERPH-408374.

Thank you for your time and effort in reviewing our manuscript and for your comments on the study.

1) In the introduction section, it is not always clear whether the impact on the quality of dialogues depends on parents' or offspring's traumatic experiences (or both). I suggest clarify this point;

We thank the reviewer for this point and have changed the text to make the impact of the individual’s trauma exposure more clear (page 1, line 43; page 2, line 48, line 64, line 66, line 73, line 84, line 85-86; page 3, line 120).

2) At line 71, the authors state "less resolved mothers...". In what sense "resolved"? Do the authors refer to an attachment theory-related resolution of loss or trauma, or do they refer to a psychoanalytic framework?

Resolved in this sense refers to the attachment theory (Bowlby, 1982) which states that resolution involves the individual’s ability to integrate the trauma in their view of the world and come to terms with the abuse, enabling them to focus on the present and the current signals of the child. This theoretical framework is explained more in the introduction (page 2, line 72-78).

3) I feel that the reader could benefit from some explanation of how different kinds of trauma (violence, sexual abuse, neglect, etc.) impact the quality of parent-child interaction and dialogues; and is there any difference in case of interpersonal trauma?

Trauma exposure impacts parenting and mutual communication (Banyard et al., 2003; Fraiberg et al., 1975) and similar interaction problems have been found for different kinds of interpersonal trauma. For example mothers who had been exposed to domestic violence with their spouse (DeVoe et al., 2002), as well as sexual abused mothers (Banyard et al., 2003) showed more aggressiveness, less warmth, less consistent parenting and less attention for the emotional needs of their children. This is stated more clearly on page 2, line 90. Other studies have shown that maternal childhood neglect has similar effects as abuse on later parent-child interaction (e.g. Bailey et al., 2012, Lang et al., 2010). However, no sample of neglected mothers was included in our current study, which is why we did not describe this relationship between neglect and parent-child interaction in the introduction. In addition, multiple forms of childhood abuse often co-occur (Dong et al., 2004), making it difficult to disentangle individual effects of single forms of abuse on outcome measures, in this case quality of parent-child interaction. We therefore deem it more appropriate to focus on the effect of interpersonal trauma in general on the quality of parent-child interaction and not pay much attention to individual specific effects. Previous studies show that interpersonal trauma has a greater impact on victim’s functioning than non-interpersonal trauma (Cromer & Smyth, 2009). In our sample only dyads who had been exposed to interpersonal trauma have been included, which is why we focused our introduction on these types of trauma and have stated this more clearly in the text where appropriate (page 1: line 15, line 20; page 2: line 68, line 78, line 83, line 88, line 89; page 3: line 99, line 104, line 105, line 125, line 138, line 144; page 4: line 162, title Table 2; page 9: line 244, line 253; page 11: line 334; page 12: line 371, line 374; page 13: line 400, line 404).

4) I suggest adding specific objectives.
The objectives of the study have been added to paragraph 1.4 (line 137, line 142-143).

5) It is not clear who administered the measures? Were the tools administered by the same professionals in the different studies the sample comes from?
In the five different studies different assessors were trained to administer the measures. Data of participants who participated in previous studies were pooled to form the sample for the current study.  The current sample consists of 32 sexually abused mothers from the study of Koren-Karie et al. (2008), 17 sexually abused children, 13 sexually abused mothers and children and 29 control dyads from the study of Van Delft et al. (2018), 30 children and their mothers exposed to interparental violence and 28 control dyads from the study of Visser et al. (2015), 121 dyads exposed to interparental violence from the study of Overbeek et al. (2012) and 29 control dyads from the study of Koren-Karie & Getzler-Yosef (2018). The content of the dialogues was coded by different coders than the quality of parent-child interaction and coherence. All coders were blind to group allocation (trauma/non-trauma). This has been added to the method-section (page 4, line 170-171).

6) The analyses were well conducted and I do not have any suggestion.
We thank the reviewer for this thoughtful compliment.

7) The result section is well written and clear.
We are happy to hear the reviewer had no additional feedback on the Results section.

8) In the discussion section, I would suggest adding a brief separate paragraph on possible clinical implications.
Thank you for this suggestion for improving our manuscript. However, since this is a non-experimental study it is difficult and highly speculative to draw direct clinical implications. We did include an additional sentence on possible clinical avenues in the conclusion (page 13: line 397-401).

Round  2

Reviewer 1 Report

Thank you for your considered response. 

I just have one remaining thought the authors might consider in relation to the use of mutually exclusive coding. In the discussion line 318-319.  “Contrary to expectation, trauma-exposed dyads did not describe fewer positive and more 319 negative stories about parental figures.” In the example given in your response, you describe a traumatic experience that features a father behaving in a very negative way, this would be coded as ‘traumatic’ and not ‘parent negative’. I acknowledge the reasoning behind this approach in the context of the regression analysis. But, could this approach to coding have artificially reduced the possible number of stories coded as parent negative for the trauma group? Given that you adopted mutually exclusive coding are you confident that there is truly no difference in the number of parent neg/pos stories? It's not a major point but one the authors might consider.   

Author Response

I just have one remaining thought the authors might consider in relation to the use of mutually exclusive coding.

In the discussion line 318-319.  “Contrary to expectation, trauma-exposed dyads did not describe fewer positive and more 319 negative stories about parental figures.” In the example given in your response, you describe a traumatic experience that features a father behaving in a very negative way, this would be coded as ‘traumatic’ and not ‘parent negative’. I acknowledge the reasoning behind this approach in the context of the regression analysis. But, could this approach to coding have artificially reduced the possible number of stories coded as parent negative for the trauma group? Given that you adopted mutually exclusive coding are you confident that there is truly no difference in the number of parent neg/pos stories? It's not a major point but one the authors might consider.   

We agree with reviewer 1, that a topic could be both related to a traumatic event as well as to a negative experience regarding the parent, although generally the emphasis in a story was on one topic. The way of mutually exclusive coding may be a possible explanation of the lack of differences found in topics about negative feelings regarding parents. We added this explanation to the discussion in lines 327-329:

“In the current study, the mutually exclusive coding of the topic of a story might have hindered coding more stories as negative about parental figures in trauma-exposed dyads. On the other hand, compared to earlier studies, mother and child participated together in a discussion task, which could make it more difficult for the child to discuss negative stories with the parent in which the parent is the focus of the negative emotions, especially when the mother-child interaction is of lower quality; and could be a sign of the child’s tendency to please the mother. Support for this interpretation can be found in the fact that for the emotion happy, trauma-exposed dyads were even more likely to discuss positive stories about parental figures.”